# Optoelectronic Instrumentation and Measurement Strategies for Optical Chemical (Bio)Sensing

**Francisco Ferrero Martín** [1,*], **Marta Valledor Llopis** [1], **Juan C. Campo Rodríguez** [1], **Alberto López Martínez** [1], **Ana Soldado Cabezuelo** [2,*], **María T. Fernández-Arguelles** [2] and **José M. Costa-Fernández** [2]

1   Department of Electrical, Electronic, Computers and Systems Engineering, University of Oviedo, 33204 Gijon, Spain; valledormarta@uniovi.es (M.V.L.); campo@uniovi.es (J.C.C.R.); UO181549@uniovi.es (A.L.M.)

2   Department of Physical and Analytical Chemistry, University of Oviedo, 33006 Oviedo, Spain; fernandezteresa@uniovi.es (M.T.F.-A.); jcostafe@uniovi.es (J.M.C.-F.)

*   Correspondence: ferrero@uniovi.es (F.F.M.); soldadoana@uniovi.es (A.S.C.);
    Tel.: +34-985-182552 (F.F.M.); +34-985-103583 (A.S.C.)

**Abstract:** There is a growing interest in the development of sensitive, portable, and low-cost instrumentation for optical chemical (bio)sensing. Such instrumentation can allow real-time decision-making for industry, farmers, and researchers. The combination of optical fiber schemes, luminescence spectroscopy techniques, and new materials for sensor immobilization has allowed the growth of optical sensors. This article focuses on the development of low-cost optoelectronic instrumentation and measurement strategies for optical chemical (bio)sensing. Most of the articles in this field have focused on the chemical sensors themselves, although few have covered the design process for optoelectronic instrumentation. This article tries to fill this gap by presenting designs for real applications, as carried out by the authors. We also offer an introduction to the optical devices and optical measurement techniques used in this field to allow a full understanding of the applications.

**Keywords:** optical fiber sensors; optoelectronic instrumentation; fluorescence; phosphorescence; reflectance; ratiometric measurements

## 1. Introduction

Currently, there is considerable interest in the development of specific, sensitive, low-cost, and portable optoelectronic instrumentation that is specially adapted to optical chemical (bio)sensing. Recent developments of novel sensitive and selective materials play important roles in the measurement of chemical and biochemical species in complex industrial, environmental, and agri-food samples. In addition, fiber optic technology is widely applied in many optical measurement processes because of its important advantages, such as noise immunity and the possibility of its use for remote and multiposition measurements.

The development and investigation of fiber optic chemical sensors (FOCS) and biosensors (FOBS) is reflected by the hundreds of studies performed in the past decade, as reported in [1–3]. The use of optical fibers in combination with new materials for use in chemical sensing, i.e., for recognition of analytes via specific molecular interactions, is being investigated for the development of new optical chemical sensors [4]. This allows the development of robust instrumentation for monitoring gases and vapors, pH, ions, organic chemicals, humidity, hydrogen peroxide, and hydrazine in areas such as the chemical industry, biotechnology, medicine, environmental sciences, industrial production monitoring, and the automotive industry [5].

On the other hand, advanced optical sensing schemes [6,7], such as ratiometric measurements [8–13], are alternatives to classical intensity or lifetime measurements [14–16] due to their proven insensitivity to background light and instrumentation fluctuations.

These techniques can be applied for the development of robust optical sensor instruments based on affordable optoelectronic components.

The basic instrumentation associated with optical sensors is simple in principle because it uses conventional, commercially affordable spectroscopic components (optical and electronic), typical light sources, optical filters or monochromators, light couplers, and light detectors, whose characteristics and costs will depend on the specific needs. The possibility of the individual selection of each of these components offers a great variety of combinations. In fact, it is possible to custom design an instrument so that it has the characteristics required for each specific case.

Figure 1 shows a block diagram of a typical instrumentation system related to optical sensors. It consists of four main blocks: light source, wavelength selector, light detector, and signal processing unit. Optical fibers, which may or may not be part of the system, are used to link the components of the optoelectronic system. The optical sensor generates a response in relationship to the analyte concentration in the sample. The signal processing will extract the information from the analytical parameter to evaluate. There is no universal solution for the design of the instrumentation system in Figure 1. The characteristics of the optical sensor force the type of measurement strategy and the requirements of optical devices. Ref. [17] reports an extensive overview of instrumentation for fiber optic chemical sensors, and [18] presents a good review of low-cost optical instrumentation for biomedical measurements.

We structure the rest of the article as follows. Section 2 reports the main characteristics of current optical devices and the selection criteria for their application in optoelectronic instrumentation. Section 3 focuses on the main spectroscopic techniques for optical luminescence sensing. Section 4 introduces the design of optoelectronic instrumentation for measuring chemical parameters by means of two representative examples. Finally, Section 5 provides conclusions.

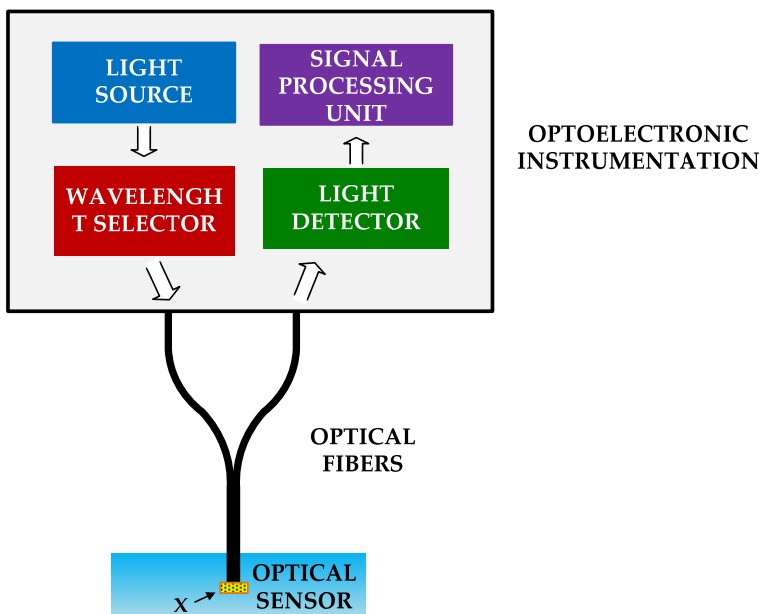

**Figure 1.** Typical components of an optoelectronic system associated with optical sensors. Here, X symbolizes the variable that modulates some parameters of the light.

## 2. Optical Devices

### 2.1. Light Sources

The light source used to excite the optical sensor must meet the requirements established for its absorption spectrum and produce an intense radiation that does not change over the life of the system. The light sources must have appropriate coupling systems with the optical fiber to ensure that the passage of light is carried out with minimal energy loss.

We also highly recommend the use of light modulated at a certain frequency to reduce the effect of external interference, which can be eliminated by filtering the output signal of the detector to leave only the frequency corresponding to the modulation of the light source. This frequency must not be so high as to compromise the dynamic limitations of the other components of the system, nor too low to be confused with signals from artificial lighting or other sources.

There are two basic lamp types available—incandescent (or filament) and gas discharge. Until a few years ago, the most common light sources were incandescent lamps. The advantage of this type of light source is its wide emission spectrum, ranging from UV to IR. On the other hand, this source has low energy efficiency, low mechanical stability, and requires an external device to modulate the light. The gas discharge lamp is composed of two electrodes (anode and cathode) and a gas of neutral atoms contained by an envelope of suitable material (glass, quartz, or synthetic quartz). For optical instrumentation, gas discharge lamps can be pulsed (flash lamp) or continuous-arc lamps. Flash lamps, primarily due to their commercially significant application as laser pumping sources, have experienced considerably more innovation in recent years than other light sources. Hamamatsu Photonics produces three types of gas discharge lamps, namely deuterium lamps, xenon and mercury–xenon lamps, and xenon flash lamps [19]. Deuterium lamps are discharge light sources that utilize the stable arc discharge of deuterium gas (D2). The distinguishing characteristic of this gas is a continuum emission from 180 to 400 nm; therefore, its principal application is as a source for UV spectroscopy.

Xenon lamps are filled with xenon gas that emits "white light" at a high color temperature of 6000 K, which is close to that of sunlight and covers a broad continuous spectrum (185 nm to 2000 nm) from the UV to IR range. These xenon lamps are ideal as light sources for various types of photometric instruments, such as spectrophotometers, liquid chromatographs, microscope light sources, color analyzers, and color scanners. Mercury–xenon lamps produce high radiant energy in the UV region due to their optimal mixture of mercury and xenon gas. These lamps possess features of both xenon gas and mercury discharge lamps. The spectral distribution includes a continuous-line spectrum ranging from UV to IR regions of the xenon gas and strong mercury line spectra in the UV to visible region. Compared to xenon lamps, the radiant spectrum in the UV region of mercury–xenon lamps is sharper in width and its peak is higher in intensity. These features make mercury–xenon lamps ideal UV light sources. Their typical applications are in fluorescent microscopy, as blood analyzers, and in UV irradiation equipment. Xenon flash lamps are pulsed light sources that emit light with an instantaneously high peak output. The emitted light is a continuous spectrum spanning from the UV to the IR region and is used for a wide range of applications, including chemical analysis and imaging. Specially designed power supplies and trigger sockets are required to obtain maximal performance from xenon flash lamps.

The development of optoelectronics has given rise to low-cost semiconductor-type light sources, mainly light-emitting diodes (LEDs) and laser diodes. LEDs are compact, energy-efficient light sources that can emit light over a wide range of wavelengths. The currently available LEDs cover the visible spectrum, a large part of the UV range (from approximately 250 nm), and part of the IR range (up to approximately 4.5 μm) [20]. The full-width half-maximum (FWHM) is usually in the range of 10 to 100 nm. The optical power is typically in the range of 1 to 170 mW. Other advantages are the possibility of direct electronic modulation at high frequencies, long life (100,000 h), small dimensions, high energy efficiency, and low cost. LEDs and fiber optics can be purchased already assembled, which guarantees perfect coupling [21].

Other light source options are laser diodes (LDs). They are available with center wavelengths in the range of 400–2000 nm and output powers from 1.5 mW up to 3 W. It should be noted that the absolute rating is of the output power, not of the drive current. Their lifetime is also longer and the price is much higher than conventional LEDs. Until recently, there were no laser diodes in the violet–blue region. The price of a laser diode in this region, at certain wavelengths, can exceed $1000. The most important factor when

choosing a LD is probably wavelength. Another highly important factor is diode packaging. The "correct" choice of packaging is dependent on the intended use and lab requirements. Laser diodes can also be modulated, although the particularities of their characteristic light–current power curve do not make them the most suitable option. They require a minimum current value to produce the laser effect. Table 1 summarizes the most outstanding features of the considered light sources.

**Table 1.** The main characteristics of light sources used for optical sensing.

| Gas Discharge Lamps | LEDs | Laser Diodes |
|---|---|---|
| <ul><li>Spectrum: UV–IR</li><li>Low mechanical stability</li><li>External device to modulate light</li><li>Specially designed power supplies and trigger sockets</li><li>Photometric instruments</li></ul> | <ul><li>Spectrum: UV–IR</li><li>High energy efficiency</li><li>Bandwidth: 10–40 nm</li><li>Optical power: 0.5–20 mW</li><li>Long life span</li><li>Direct electronic modulation</li></ul> | <ul><li>Spectrum: Visible</li><li>Bandwidth: 0.5–3 nm</li><li>Optical power: 1.5–3 W</li><li>Long life span</li><li>Direct modulation (requires minimum current)</li><li>Specially designed power supplies</li></ul> |
|  |  |  |

## 2.2. Light Detectors

A light detector (photodetector) transforms the optical signal into an electrical signal, preserving the original chemical information. The spectral characteristics of the photodetector must be adapted to the emission spectrum of the optical sensor to avoid a loss of information at critical wavelengths. In addition, the photodetector system must have a high sensitivity to ensure a high signal-to-noise ratio (SNR).

The most used photodetectors in instrumentation are photodiodes and photomultiplier tubes (PMTs) [22]. Photodiodes (PDs) are the preferred detectors for low-cost instrumentation. They can operate at high levels of light without degradation. Depending on the semiconductor material, their spectral responses vary between 180 and 2600 nm. PDs are fast (depending on the internal capacitance, the bandwidth in some cases is up to 1 GHz), robust, and cheap, and in photovoltaic mode they do not require power. Their main drawbacks are their relatively high levels of noise and their lack of internal amplification. As their output signal is usually small, they require an additional amplifier, which also introduces noise. PDs can be used with low light intensities, limiting the signal bandwidth, and with noise cancellation techniques. Silicon PDs are appropriate for high-speed applications, such as for spectrophotometry, optical measurement equipment, analytical instrument, and radiation detection. InGaAs PDs used for near-infrared light detection are suitable for a wide range of applications, including optical communication, analysis, and measurement. UV LEDs are well suited for spectroscopic applications in instrumentation used for analytical and life sciences.

Avalanche photodiodes (APDs) are devices that require a high reverse voltage (between 100 V and several kV), resulting in internal amplification due to the phenomenon of avalanche multiplication. APDs have a higher SNR than silicon photodiodes, as well as a fast response rate, low dark current, and high sensitivity. Their spectral range is between 200 and 1150 nm. In the non-avalanche mode, one can achieve internal amplification rates of between 10 and 100 times those of a PD. APDs are fast; they can be used at frequencies up to 10 GHz. Although their internal noise is higher than normal and more expensive

PDs, they are becoming increasingly popular because they can tolerate intense lighting (optical power meters) and their sensitivity is comparable to some PMTs.

PMTs are the best choice when radiant sensitivity, speed, and minimum noise are the main requirements of the system. The internal amplification of a PMT is up to $10^6$ or more. Another advantage is the possibility of larger areas of detection. The noise is significantly lower compared to solid-state devices. The drawbacks of PMTs are the need for a high-voltage power supply (600–1200 V) and the possibility of destruction by overexposure. As such devices are typically made of glass, they have low mechanical stability; however, new miniature PMT designs (with metal housings, built-in power supply, and current–voltage converters included) make their use in instrumentation increasingly common in cases where the cost of the equipment is not a limitation. PMTs are widely used in many applications, including in industry, medicine, and academic research, where high-sensitivity photodetectors are required. Typical applications are in flow cytometers, blood inspection, hygiene monitors, portable survey meters, environmental measurement, and semiconductor wafer inspection systems.

In summary, to measure optical radiation, the first consideration is the sensitivity of the detector in the band of interest, defined as the electric current generated by the optical power (A/W). Of course, each detector has its own peculiarities; for example, some detectors require cooling, for others it is necessary to modulate the radiation or they require a high voltage, while others can be damaged if subjected to excessive radiation or their sensitive window is touched. Each of these peculiarities must be evaluated when selecting the best option. Hamamatsu Photonics provides a guide for the selection of light detectors [23]. Figure 2 shows the wavelength ranges covered by the discussed light sources and detectors. Table 2 summarizes the main characteristics of the light detectors considered for chemical sensing.

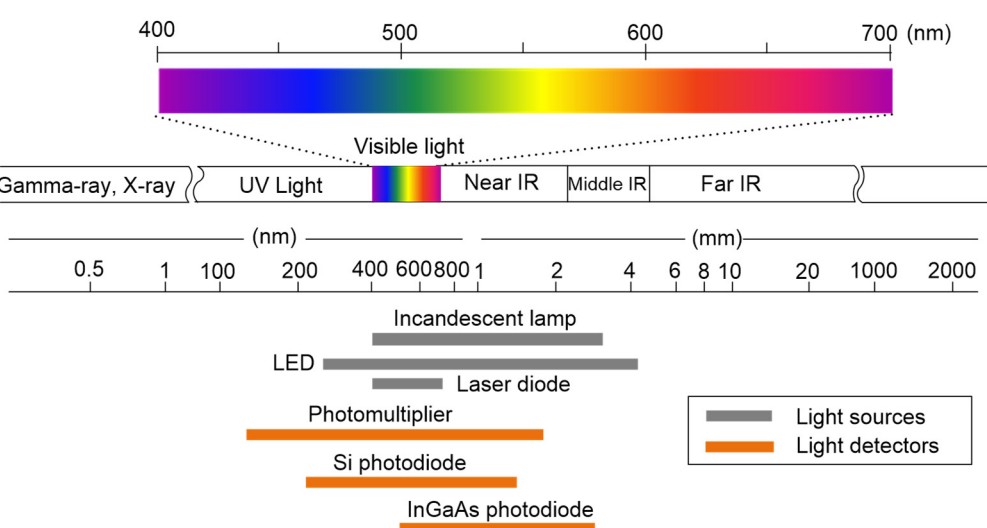

**Figure 2.** Wavelength ranges covered by various light sources and light detectors.

**Table 2.** The main characteristics of light detectors used for optical sensing.

| Photodiodes | Avalanche Photodiodes | Photomultipliers |
|---|---|---|
| • Range: UV–NIR<br>• Fast and robust<br>• Low cost<br>• Relatively high noise<br>• Requires additional amplification | • Range: UV–NIR<br>• High sensitivity and speed<br>• Reverse high voltage<br>• Internal amplification<br>• Low light levels | • Large spectral range<br>• High voltage supply<br>• Very high internal amplification<br>• Low noise<br>• High cost<br>• Low mechanical stability [24] |

### 2.3. Wavelength Selector

In the design of optoelectronic systems, it is common to use a device that allows the selection of certain wavelengths. The most common devices used to perform this function are optical filters and monochromators. An optical filter located between the optical fiber and the emission light detector can greatly reduce the signal provided by the reflection of the excitation and the signal produced by light interference. The optical filters used in spectroscopy are generally interference-based. They are used to monitor light at certain wavelengths and are based on the arrangement of thin layers of dielectrics that produce interference between the wavefronts, resulting in very small bandwidths. The advancement of thin-film deposition technology has allowed for filters with a bandwidth of 5 nm (FWHM), central wavelengths (CWL) between 250 and 1500 nm, transmission >90% (at the peak of the central wavelength), and optical density > 4. These filters are available as components [25] or mounted directly on photodiodes [26].

Other wavelength selection devices include monochromators, although their use is in laboratory instrumentation. They are based on the use of a diffraction network, which depending on the position, can diffract a polychromatic light source into its monochromatic components. In diode-matrix-based monochromators or CCDs, the spectrum is directed to a linear array of photodetectors. The size of the detector determines the spectral resolution. These monochromators are significantly cheaper, have no moving parts, and are very reliable. Other advantages are the simultaneous access to the entire spectrum and the integrated optoelectronic conversion; however, their resolution is lower compared to their diffraction network counterparts. Fiber optic instrumentation systems may also require the use of optical couplers or beam splitters to efficiently distinguish and separate incident and return radiation from traveling in a single fiber. Optical refraction and reflection components, such as lenses and mirrors, are also required in instruments to manipulate light and focus it more effectively on fiber optics. Table 3 summarizes the main characteristics of the wavelength selectors considered for optical chemical (bio)sensing.

**Table 3.** Wavelength selector devices used for optical chemical (bio)sensing.

| Optical Filters | Monochromators |
| --- | --- |
| • Central wavelength (CWL): 250–1500 nm <br> • Bandwidth (FWHM): 5.0–10 nm <br> • Fast and robust <br> • Low cost | • Range: 180–240 µm interchangeable grid <br> • Laboratory applications <br> • High resolution: 0.1 nm |
| 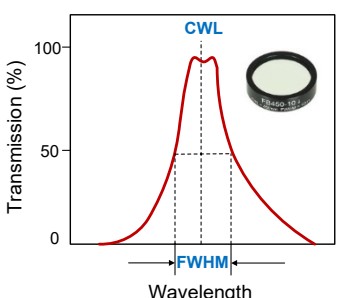 | 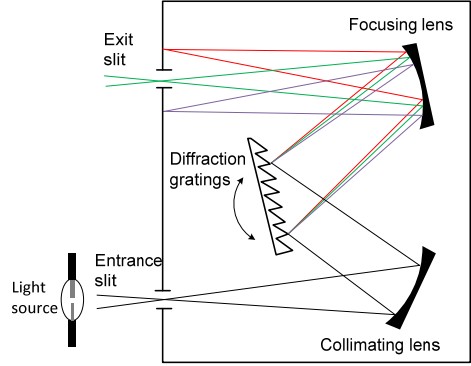 |

### 2.4. Optical Fibers

The incorporation of optical fibers in the analytical system provides attractive characteristics, enhancing the advantages offered by optical methods of chemical measurement. The ability of fibers to transmit optical signals over long distances with little power loss makes direct measurement possible in places far from the instrumentation system. The flexibility of optical fibers allows chemical analyses to be performed in hard-to-reach places. Fiber optics provide a waveguide for radiation that interrogates the molecular recognition element of the optical sensor.

Optical fibers, as shown in Figure 3, consist of a transparent dielectric cylinder (core) surrounded by a cladding of another dielectric material with a refractive index lower than that of the core. These two cylinders are protected by a coating. The propagation of light through the length of an optical fiber takes place through the phenomenon of total internal reflection. Any ray that enters through one end of the fiber into the acceptance cone will propagate through the core of the fiber. The maximum acceptance angle depends only on the refractive indices of the core ($n_1$) and the cover ($n_2$), via the following expression:

$$sen\ \theta_a = \left( n_1^2 - n_2^2 \right)^{1/2} \tag{1}$$

The light-gathering ability of the fiber is called the numerical aperture (*NA*) and corresponds to the sine of the acceptance angle:

$$NA = sen\ \theta_a \tag{2}$$

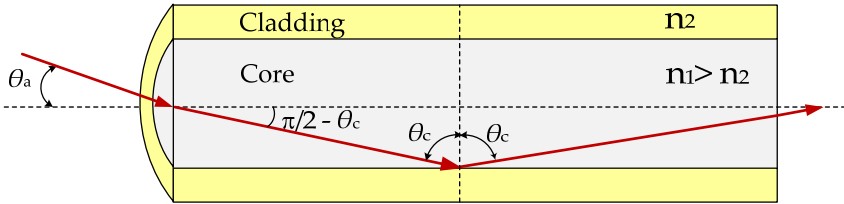

**Figure 3.** Transmission of a ray of light inside an optical fiber.

This value is of great importance for the coupling of light to optical fiber conductors. The greater the amount of light that the fiber can accept, the greater the transmission distance that can be reached, although if it is very large the system bandwidth degrades.

More details of the propagation of light in optical fibers can be found in [27,28]. The transmission of light through a fiber depends on the wavelength of the radiation and the angle of reflection. Only certain combinations (modes) of wavelengths and reflection angles are allowed. Each mode corresponds to a single angle of reflection for each radiation wavelength. For a particular wavelength, the number of propagation modes decreases with the diameter of the fiber. Optical fibers with core diameters between 2 and 8 μm allow a single propagation mode (single-mode fibers). Fibers with larger diameters, typically between 50 and 200 μm, are multimode and allow simultaneous transmission of several wavelengths through the fiber; they are typically used over shorter distances. Multimode optical fibers are commonly used in FOCS because they are easier to connect and their tolerance for lower-precision components is higher due to the large size of their cores.

The propagation of modes in an optical fiber depends on the profile of the refractive index of the nucleus. There are two types of multimode optical fibers:

- Step-index multimode optical fibers, in which the refractive index is constant;
- Graded-index multimode optical fibers, in which the refractive index varies in parabolic shape from a maximum on the conductor axis to a minimum on the coating. The modal dispersion of this type of fiber is lower, although it is more expensive.

The material used in fiber optic cables can be plastic, glass, or a combination of both glass for the core and plastic for the cover. Plastic fibers are used in the visible region of the spectrum, are lower in cost, and are easier to work with. Glass core fibers are used in the UV–visible and visible–NIR spectrum regions. They can be single-mode, multimode with a step index (such as polymer cladding silica (PCS) fibers) or gradient index, or fiber bundles made from such fibers [29]. Table 4 shows the main characteristics of optical fibers used for (bio)sensing applications. Multimode optical fibers are commonly used in FOCS.

**Table 4.** The main characteristics of optical fibers.

| Single-Mode Fiber | Graded-Index Multimode Fiber | Step-Index Multimode Fiber |
|---|---|---|
| <ul><li>$\varnothing_{core}$: 2–8 μm</li><li>Requires laser</li><li>High bandwidth</li><li>Difficult to manipulate</li><li>Communications</li></ul> | <ul><li>$\varnothing_{core}$: 50–200 μm</li><li>Simpler</li><li>Laser or LED</li><li>Shorter distances</li></ul> | <ul><li>$\varnothing_{core}$: 50–200 μm</li><li>Laser or LED</li><li>Lower modal dispersion</li><li>Higher cost</li></ul> |
|  |  |  |

There are several schemes to connect optical fibers and sensing areas, as shown in Figure 4. The instrument shown in Figure 4a uses a single fiber or a fiber bundle. The instrument has a beam splitter to isolate the returning radiation and focus it on the photodetector. Figure 4b shows a bifurcated optical fiber. This scheme requires simple instrumentation and is highly efficient at collecting light. Figure 4c shows that the sensing element is deposited over the core fiber, acting as the cladding. The evanescent wave interacts with the sensing area and is attenuated according to the Lambert–Beer law. This schema requires a thin reagent layer and the response time is short.

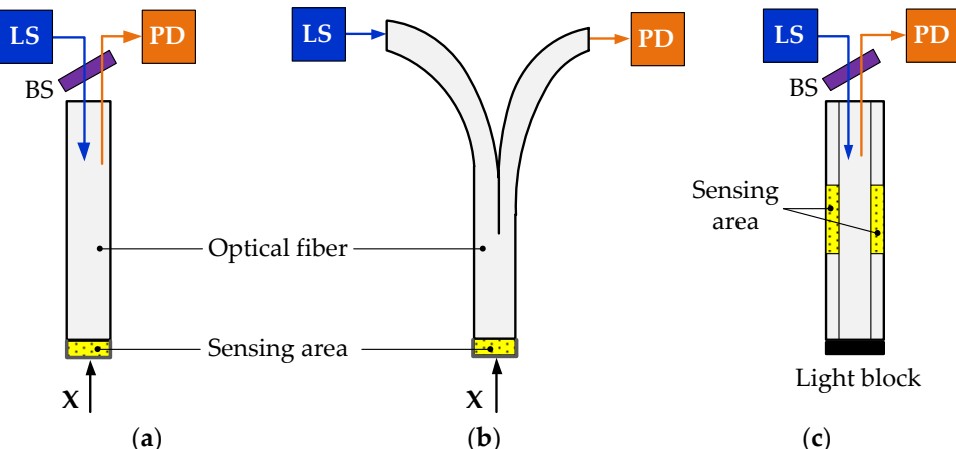

**Figure 4.** Typical schemes used for chemical sensors and biosensors: (**a**) single fiber (or bundle); (**b**) bifurcated optical fiber sensor; (**c**) evanescent wave sensor. LS = light source; PD = photodetector, BS = beam splitter; X = parameter of interest.

Special optical fibers must be used for harsh environments [30]. The most common parameters are temperature and strain or stress sensing, although other parameters such as the pressure, magnetic field, voltage, and chemical species can also be measured. The coating of the optical fiber plays the key role in protecting the glass. If the coating survives, the glass will continue to perform. Special coatings such as polyimide, silicone, and high-temperature acrylate are available that are suitable for higher temperatures. In high-pressure ambient environments, some polyimide coatings may degrade and peel off from the glass, exposing the surface. A number of added functionalities can be obtained by replacing the polymer coating with metal. Unlike polymers, metals do not outgas in vacuums, do not ignite, and they lend themselves to mounting by soldering. Regarding harsh environments, metal-coated fibers also exhibit many attractive features. As with carbon-coated fibers, metal-coated fibers are protected against ingression of water and hydrogen. Furthermore, metal coatings provide unsurpassed heat resistance; fibers coated with aluminum or copper, for example, may operate at temperatures well above 400 °C. fiber Bragg grating temperature sensors have also been used nuclear environments due to the presence of ionizing radiation fields [31], as well as in harsh aerospace environment [32]; however, only in recent years has this technology matured sufficiently to allow real field applications.

*2.5. Optical Sensors*

As shown in Figure 5, biochemical sensors sensitive to specific chemical species are composed of two parts: (a) a "receptor" or molecular recognition element capable of interacting selectively with the species of interest (analyte), being the result of a physical or chemical change (e.g., heat, electron transfer, absorption or emission of light, or vibration) of the system, the intensity of which will be related to the concentration of the species to be analyzed; (b) a "physical transducer", i.e., a measurement zone in which the change is transformed into a measurable physical signal. According to the nature of the physical signal that is generated and measured, i.e., the type of transducer, the sensors for biochemical species can be classified as electrochemical, optical, thermal, or piezoelectric. Optical sensors are chemical sensors that provide an optical response depending on the concentration of the analyte in the sample. They can be classified according to the optical property that has been measured, e.g., absorbance, reflectance, fluorescence, phosphorescence, or Raman dispersion. We will introduce these optical properties in Section 3.

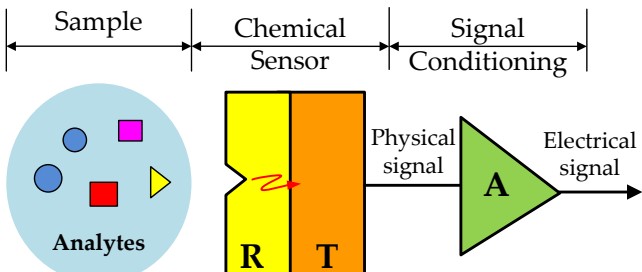

**Figure 5.** Basic constitution of a (bio)chemical sensor sensitive to a specific analyte: R = receptor (molecular recognition element); T = transducer; A = amplifier.

Optical sensors have numerous advantages over conventional sensors, such as their selectivity, immunity to electromagnetic interference, and safety while working with flammable and explosive compounds. They are also sensitive, cheap, and non-destructive. In addition, multiple analyses can be easily miniaturized and allowed with a single control instrument at a central site; however, optical sensors also have some disadvantages, namely that ambient light can interfere with their operation, their long-term stability is limited due to indicator leaching, their dynamic range is limited, and loss of selectivity occurs after immobilization.

Fiber optic chemical sensors (FOCS) are a subclass of optical sensors in which an optical fiber is employed to transmit the electromagnetic radiation to and from a sensing region that is in direct contact with the sample. The spectroscopically detectable optical property can be measured through the fiber optic arrangement, which allows remote and inexpensive sensing systems.

The production of cheap and high-quality optical fibers (initially conceived for application in the field of telecommunications) has opened new frontiers in chemical and clinical analysis with the introduction of biochemical fiber optic sensors. In particular, the inherent properties of optical fibers (small size and weight, flexibility, thermal and chemical stability, biocompatibility, and immunity to electrical interference) make them ideal radiation conductors for manufacturing optical sensors.

There are two large types of FOCS, also called optrodes, which highlight the idea that their use is similar that of electrodes, although the operation principles are different:

(1)  Extrinsic sensors use fiber optics only as a means of transmitting light from the light source to the sensitive area and from this to the photodetector. We refer to this type of sensor in this article;

(2)  Intrinsic sensors use fiber optics as a light guide and as a transducer. The variable to be measured modifies certain properties of the fiber, such as the refractive index or the absorption coefficient. These sensors can use interferometric configurations, fiber Bragg grating (FBG), long-period fiber grating (LPFG), or special fibers (doped fibers) designed to be sensitive to specific perturbations. These types of sensors are commonly used as physical sensors (e.g., pressure and temperature gauges), although their applicability for biochemical species is restricted.

In [1], Wang and Wolfbeis present a complete review of FOCS studies published between October 2015 and October 2019. The main applications for FOCS are in the sensing gases and vapors, medical and chemical analysis, molecular biotechnology, marine and environmental analysis, industrial production monitoring, and bioprocess control. Many chemical analytes can be sensed. These range from various gaseous species ($NH_3$, $H_2$, $CH_4$, $H_2S$, $CO_2$, $NO_2$, $O_2$) to volatile organic compounds (such as ethanol, methanol, acetone, toluene, and formaldehyde), as well as heavy metal ions such as $Hg^{2+}$, $Pb^{2+}$, $Mg^{2+}$, $Cd^{2+}$, $Ni^{2+}$, and $Mn^{2+}$. Other applications are for sensing of temperature and relative humidity (RH), water fractions in organic solvents, pH values, ions ($Mn^{2+}$, $Cd^{2+}$, $Pb^{2+}$, $Hg^{2+}$, $Zn^{2+}$), and organic species (glucose, fructose, oils, etc.).

## 3. Spectroscopic Techniques

The combination of optical fibers, spectroscopy techniques, and new materials for sensor immobilization has improved the development of optical sensors [33–35]. All of these techniques rely on the interactions of light with matter. Such interactions produce a series of optical phenomena that can be used to quantify the characteristics of matter, resulting in optical sensors. The wavelength range covered by the optical methods applied in fiber optic sensors can be between the UV region (above 200 nm) and the near-infrared region (below 3 μm). The most significant types of interaction of light with matter are absorption, reflection, scattering, and fluorescence, as shown in Figure 6. If the matter is excited with light of a certain wavelength, $\lambda_{exc}$, one part of the light is absorbed, another part is scattered, and another part is transmitted with the original wavelength because there are no changes in the energy levels of the electrons. On the other hand, the absorbed light can produce changes in the electron levels of certain molecules, causing a new emission of light (luminescence) of lower energy and of greater wavelength, $\lambda_{emi}$, than the original one.

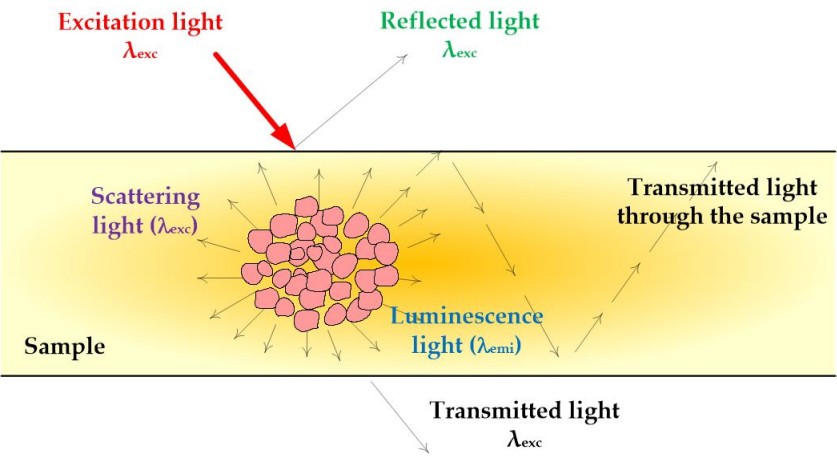

**Figure 6.** Optical phenomena due to light–matter interaction.

### 3.1. Absorbance, Reflectance, and Luminescence

Radiation absorption in the UV–visible region is one of the most popular methods in conventional analytical chemistry. When a ray of light of a certain wavelength and intensity ($I_0$) comes into contact with the dissolution of a chemical compound, the compound will absorb some of the radiation from the light ($I_a$). The remaining light ($I$) will pass through the solution and reach the light detector. It is true that $I_0 = I_a + I$. Absorbance indicates the amount of light absorbed by the sample. The absorbance coefficient is defined as:

$$A_\lambda = log_{10}(I_0/I) \tag{3}$$

The Lambert–Beer law defines the relationship between absorbance and analyte concentration in the solution ($C$, mol L$^{-1}$) as:

$$A_\lambda = log_{10}(I_0/I) = \varepsilon \cdot l \cdot C \tag{4}$$

where $\varepsilon$ is the molar absorption coefficient (L mol$^{-1}$ cm$^{-1}$) and $l$ the length of the optical path in the cuvette containing the sample (cm). It is important to note that the molar absorption coefficient is a function of the wavelength; therefore, the Lambert–Beer law is only true at a single wavelength (monochromatic light). The identification and quantification of a substance in a solution is carried out from the analysis of the absorption spectrum, which is a graphical representation indicating the amount of light absorbed ($\varepsilon$) at different wavelengths. A detailed study of this spectrum is required because in the solution there may be more than one light-absorbing substance.

When a chemical sensor is in an optically opaque medium or one with a weak ability to transmit light, the intensity of the reflected light can be used to measure the concentration of that substance. The reflection of light can be specular or diffuse. Specular reflection (SR) occurs on the surface of the medium without any transmission mediation and is governed by Snell's law. In diffuse reflection (DR), the light penetrates into the medium and reappears on the surface after suffering partial absorption and a series of multiple dispersions. The reflectance coefficient for a given wavelength is defined as:

$$\rho_\lambda = \frac{I_0}{I_{DR}} \tag{5}$$

This coefficient depends on obvious physical parameters and includes information on the quantity of a specific substance; therefore, reflectance can be used as an instrumentation parameter in the design of a sensor for that substance. As with absorbance, the identification and quantification of a substance is achieved through the analysis of the reflected light spectrum (reflectance spectroscopy). The processing of the spectrum to obtain the content information for a substance is complex, and as in the case of absorbance, mathematical and statistical methods are used. Fluorescence and phosphorescence are two phenomena involving photoluminescent molecules, consisting of the emission of light from excited electronic states of the molecule. To explain these phenomena, it is common to use to a diagram of the energy levels of a photoluminescent molecule, as shown in Figure 7, known as the Jablonski diagram [36]. A brief explanation of this diagram can help explain the difference between fluorescence and phosphorescence processes.

When a molecule absorbs UV or visible radiation, the electrons pass from the *singulete* fundamental state, $S_0$, to a vibrational state of higher energy, $S_1$ or $S_2$. From these states, the molecules can return to their fundamental state, without radiation emission or with emission; this emission is known as fluorescence. The fluorescent process, characterized by the total time between excitation from the fundamental state and return, is very short (ns). Molecules can also return to the ground state via another less usual route (low probability of occurrence), through the process called cross-system crossing. This process brings molecules to a lower energy *triplet* electronic state, $T_1$ or $T_2$, from which molecules can return to their fundamental state by emitting radiation (phosphorescence).

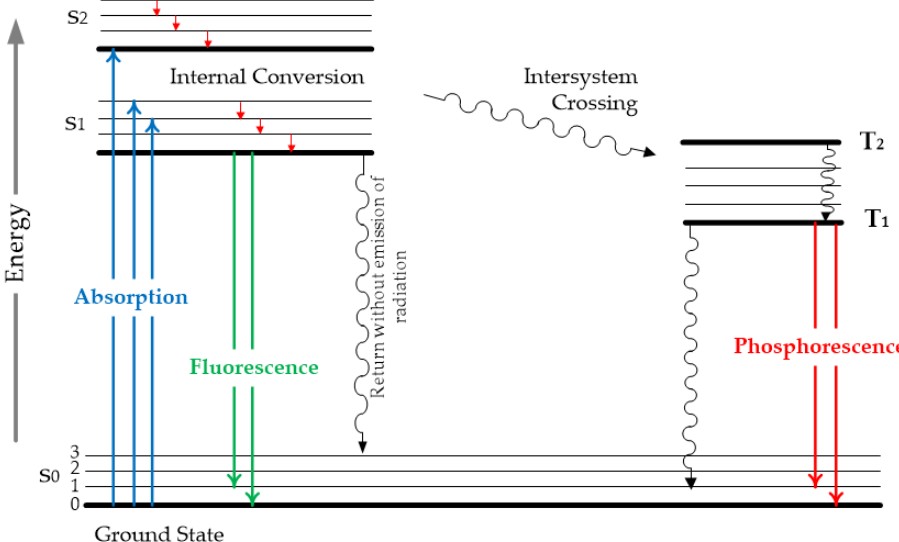

**Figure 7.** Jablonski diagram used to explain the luminescence processes.

The phosphorescent phenomenon involves much longer durations than fluorescence, ranging between 1 µs and 10 s. In addition, due to its low probability, the total intensity of radiation is low compared to fluorescence. For both fluorescence and phosphorescence, the wavelength of the emissions is greater than the wavelength of excitation (incident light),

because energy and wavelength are inversely proportional. When performing instrumentation measurements to quantify phosphorescence, it is necessary to optimize (1) the delay time required before starting the measurement to remove residual luminescence (both from the light source and from fluorescence) and (2) the integration time at which the phosphorescence signal is measured. Table 5 compares the fluorescence and phosphorescence phenomena.

**Table 5.** Comparison of fluorescence and phosphorescence phenomena.

| Feature | Fluorescence | Phosphorescence |
|:---:|:---|:---|
| Probability of happening | ● High | ● Low |
| Emission wavelength | ● Close to the excitation | ● Separate from excitation |
| Time in the excited state | ● Short (2–20 ns) | ● Long (1 µs–10 s) |
| Light intensity | ● Relatively high | ● Low |

*3.2. Measurement of Luminescence Intensity*

The measurement of luminescence (fluorescence and phosphorescence) intensity is a simple technique that can be used with all of the optical phenomena we have mentioned. It consists of measuring the light signal that is produced in a permanent regime. For weakly absorbent samples, the intensity of luminescence, $I$, is a fraction of the intensity of the excitation light, $I_{ex}$, and obeys Parker's law [13]:

$$I = (2.3)\, k\, I_{ex}\, \phi\, \epsilon_\lambda\, l\, C \tag{6}$$

where $\phi$ is the quantum yield of the luminophore, $\epsilon_\lambda$ is the molar absorbance at $\lambda_{ex}$, $l$ is the length of penetration into the sample, $C$ is the concentration of the luminophore, and $k$ a geometric factor, which depends on the configuration of the measurement system.

Figure 8 shows the instrumentation setup of a fiber optic sensor based on the measurement of luminescence intensity. An LED is commonly used as the light source if the chosen LED is capable of emitting at the appropriate wavelengths to excite the optical sensor. It is also common to modulate the light intensity at a certain frequency to reduce the effect of external interference. Another option is to use a laser diode, which can also be modulated, although the particularities of the characteristic curve do not make laser diodes the most suitable for this purpose. The optical filter between the fiber end and the light detector can improve the S/N ratio by reducing the presence of external light sources. The light detector must respond to the wavelengths selected by the optical filter. The choice depends mainly on the light levels. If these are very low, a multiplier tube should be used if price is not a limiting factor. As an alternative at lower cost, avalanche photodiodes can be used, which have internal gain, although are dependent on temperature.

The light intensity measurements can be disturbed by external factors such as fluctuations in the light source, coupling of optical fibers, changes in fiber attenuation due to curvature, and the length of the optical path [37,38]. Despite these drawbacks, intensity measurements are still used successfully to study the structure and molecular dynamics of biological and environmental sensors due to their high sensitivity and selectivity [39]. In addition, according to the application, this technique can be more versatile than other techniques, such as in the measurement of lifetimes, because sensors that exhibit changes in intensity against such parameters do not induce changes in lifetimes.

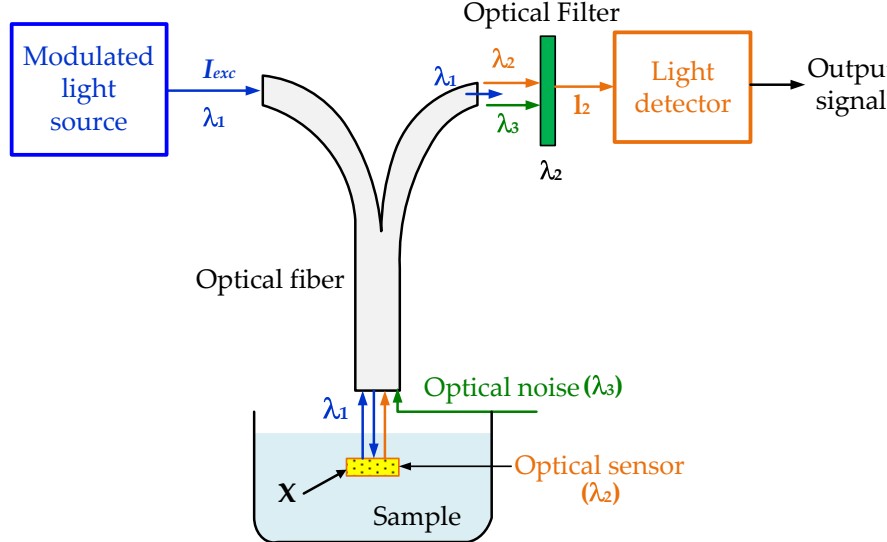

**Figure 8.** Instrumentation setup for an extrinsic optical fiber sensor used for measuring light intensity.

### 3.3. Measurement of Luminescence Lifetimes

Unfortunately, the direct measurement of luminescence intensity can be affected by a series of fluctuations that are independent of the analyte, which means that this approach must sometimes be replaced by other techniques that provide greater reliability for chemical measurements. One possibility is to measure the lifetime of the luminescent emission. This technique is conditioned by the availability of chemical sensors that undergo a change in the lifetime after an interaction with the analyte This technique is conditioned by the availability of chemical sensors that undergo a change in the lifetime after an interaction with the analyte. An additional limitation is the complexity of the measurement system, particularly for the measurement of very small lifetimes (ns).

Equation (6) describes only the permanent luminescence regime. If the excitation light is an impulse function, the luminescence decays according to:

$$I = I_0 \cdot e^{(-t/\tau(X))} \tag{7}$$

where $I_0$ is the initial intensity of luminescence ($t = 0$) and $\tau$ is the time constant, called the luminescence lifetime. It is possible to use the lifetime as a parameter to identify the concentration of a given substance provided that that substance, although not luminescent, attenuates the emission of the luminescent indicator (luminophore). In this type of sensor, the analyte reacts with the luminescent indicator, attenuating the luminescent emission, as shown in Figure 9 [40].

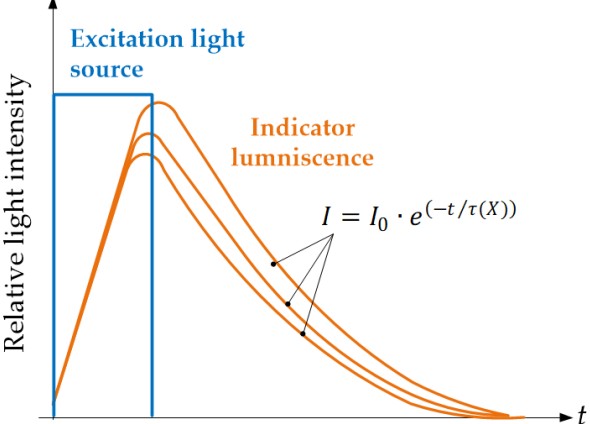

**Figure 9.** Dynamic response of a luminescent sensor to a pulse of light.

This mechanism has been used to measure very low concentrations of oxygen, halides, and various metals [41–43]. Lifetime data are independent of the luminescent sensor concentrations and changes in excitation light, sensor size, geometry, or other more complex chemical phenomena. The lifetimes of the indicators must be long enough (μs) so that they can be measured with low-cost instrumentation. An additional complication occurs when the decay times of the luminescence do not fit a monoexponential, whereby more processing will be required to achieve a better fit. Once the lifetimes are known, the concentration of the analyte can be calculated using the well-known Stern–Volmer (SV) equation [36], which linearly relates lifetimes to concentrations:

$$\frac{\tau_0}{\tau} = 1 + K_{SV}[C] \tag{8}$$

where $K_{SV}$ is the $SV$ constant, $\tau_0$ is the remaining lifetime in the absence of the quencher, and $[C]$ is the analyte concentration. Dynamic attenuation affects the lifetimes of excited states, being very useful for measuring substances that although are not luminescent, inhibit the luminescence of a chemical indicator. The measurement of the lifetime can be performed in the frequency domain or in the time domain.

### 3.3.1. Measurement of Lifetime in the Frequency Domain

The measurement of lifetimes in the frequency domain involves exciting the luminescent indicator with a sinusoidally modulated light source, typically at frequencies between 0.1 and 10 $\tau^{-1}$. Once the sensor is excited, its emission follows the excitation frequency, with a phase delay that is a function of the lifetime, as shown in Figure 10.

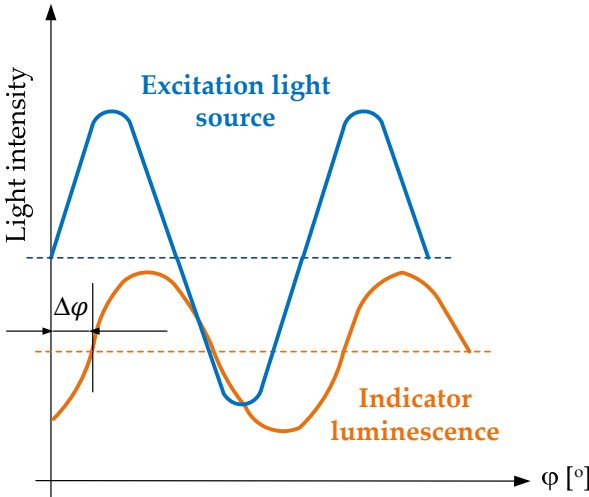

**Figure 10.** Basis of the method used for measuring the lifetime in the frequency domain.

The lifetime can be determined from the measurement of the phase delay between the excitation and light emission:

$$\tau = \frac{1}{2\pi f_{exc}} tan\varphi \tag{9}$$

where $f_{exc}$ is the frequency of the modulated excitation source. This method is relatively simple and does not require overly complex instrumentation for lifetimes ranging between μs and ms. If the indicators have ns lifetimes, higher modulation frequency light sources are required, which are more expensive. The main advantage of performing measurements in the frequency domain is that the bandwidth of the measurement system can be very small, since the measurement is carried out at a single frequency. Noise increases with bandwidth, so working under these conditions significantly improves the signal-to-noise ratio. In practice, the phase delay and the determination of the lifetime can be affected if part of the excitation light reaches the detector. This may also happen if there are other

fluorescent processes in the sensor phase, resulting in a spectral overlap between emissions. In both cases, the use of optical filters is essential. The most popular instruments for this technique are dissolved oxygen sensors [44].

### 3.3.2. Measurement of Lifetime in the Time Domain

The measurement of lifetimes in the time domain consists of applying a pulse of light of very short duration to the sample, after which a decay curve is obtained that must be processed to obtain the lifetime. Analysis in the time domain is more complex than analysis in the frequency domain [45] due to the difficulty of extracting the time constant from an exponential setting [46]. One of the methods used for measuring lifetimes in the time domain is called rapid lifetime determination [RLD] [47], as shown in Figure 11.

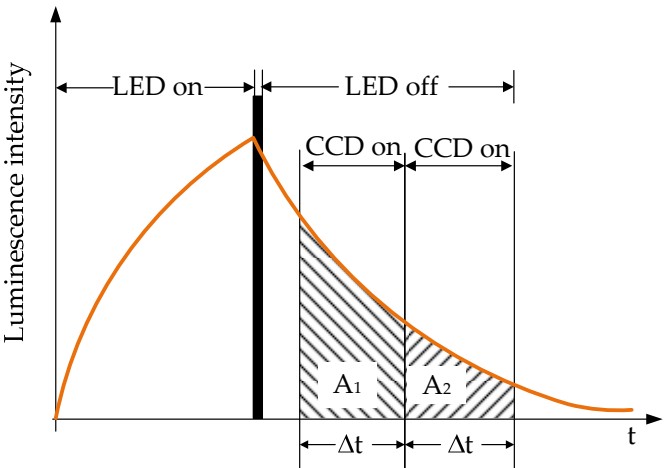

**Figure 11.** Method used for the rapid determination of the lifetime.

This method consists of measuring the areas $A_1$ and $A_2$ during the decay time of the luminescence and after the application of the excitation pulse. Assuming a monoexponential decay and identical length of the measuring windows, $\Delta t$, the lifetime can be calculated by the expression:

$$\tau = \Delta t / ln(A_1 / A_2) \tag{10}$$

where $A_1$, $A_2$ are the areas of intensity in the time increments. This scheme offers the advantage of eliminating the background luminescence, which is within the first 100 ns, before activating the CCD.

### 3.3.3. Ratiometric Techniques

The objective of ratiometric techniques is to make the result independent of the main sources of error. This independence is achieved by quantifying at least two parameters proportional to the disturbance to be eliminated. This perturbation will be canceled out when it appears both in the numerator and in the denominator of the quotient that defines the ratiometric expression. Ratiometric expressions can be applied to sensors that show changes in intensity, lifetimes, or both. This gives ratiometric techniques great versatility, as they can bring together the advantages of classical luminescent methods based on the intensity and lifetime measurements.

The most common approach in ratiometric intensity measurement is based on the ratio between the intensity levels of two emission peaks at a single excitation wavelength [36]; however, this method can suffer from photobleaching of either the indicator or the reference dye, causing changes in the measured intensity ratio. An alternative to this well-established ratiometric method is shown in Figure 12. This alternative is based on the use of a single emitter and the simultaneous measurement of the fluorescence emission and reflectance intensity from the dye. The scheme is similar to that shown in Figure 8, except that a trifurcated optical fiber and two photodetectors are used—one for sensor emission and one

for specular reflectance of the excitation signal. Because the specular reflectance does not depend on the analyte, as it does not interact with the sensor, it can be used as a reference signal. The perturbation will be canceled when it appears in both the numerator and the quotient denominator of the following ratiometric expression:

$$R = \frac{I_{\text{lum}}}{I_{\text{ref}}} \approx \frac{k_1 I_{\text{ex}}[\text{Ind}]}{k_2 I_{\text{exc}}} = \frac{k_1}{k_2}[\text{Ind}] \tag{11}$$

where $I_{\text{lum}}$ is the intensity of luminescence and $I_{\text{ref}}$ the intensity of the specular reflectance. Table 6 summarizes the advantages and disadvantages of common luminescence spectroscopy techniques.

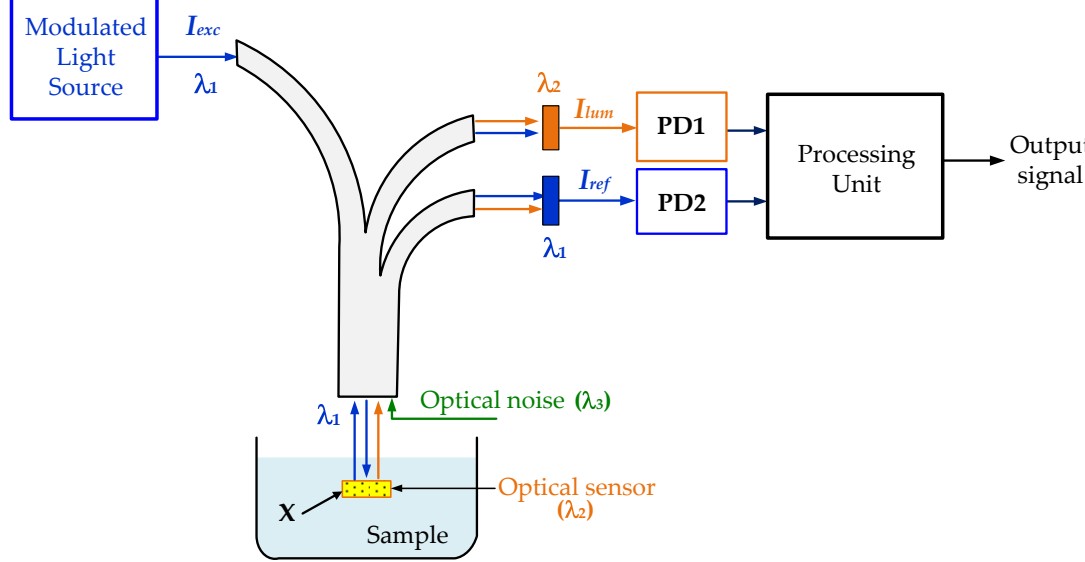

**Figure 12.** Instrumentation setup for a fiber optic sensor based on the ratiometric measurement of intensity. PD = photodetector.

**Table 6.** Advantages and disadvantages of common luminescence spectroscopy techniques.

| Spectroscopy Technique | Advantages | Disadvantages |
|---|---|---|
| Measurement of light intensity | • Simple<br>• Accurate in the laboratory | • Fluctuations of the excitation light<br>• Light losses in the optical path<br>• Photobleaching of the indicator |
| Ratiometric intensity measurement | • Relatively simple<br>• Can cancel variations in the indicator concentration, geometry, and source intensity | • Background fluorescence<br>• Light scattering and reflections are not compensated |
| Measurement of lifetime in the time domain | • Relatively simple<br>• Background fluorescence can be eliminated | • High-speed electronics<br>• Fast-pulsed optical sources |
| Measurement of lifetime in the frequency domain | • Very simple but quite accurate results<br>• Low-cost, high-brightness optical sources<br>• Standard photodetectors | • More expensive light sources for lifetimes in the range of ns |

## 4. Design of Applications

To put into practice the concepts we have discussed, in this section we provide two design examples for optoelectronic instrumentation. Although the measurement of each parameter has its own particularities, it can be stated that the design flow for this kind of instrumentation consists of five basic stages, as shown in Figure 13. Logically, the first task is to synthesize the optical sensor. This process is outside the scope of our article. The second stage is the optical characterization of the sensor phase. For this purpose, a reference piece of equipment such as a fluorescent spectrometer should be used. The results from this second stage are the excitation and emission spectra. Based on the particularities of the spectra, a specific measurement technique can be selected. The fourth stage involves the design of the measurement system, in which optical devices have to be selected. Depending on the specific application, the use of an optical fiber can be justified. The final stage is to obtain the experimental results, including the calibration curve, limit of detection, measurement margin, sensitivity, and repeatability.

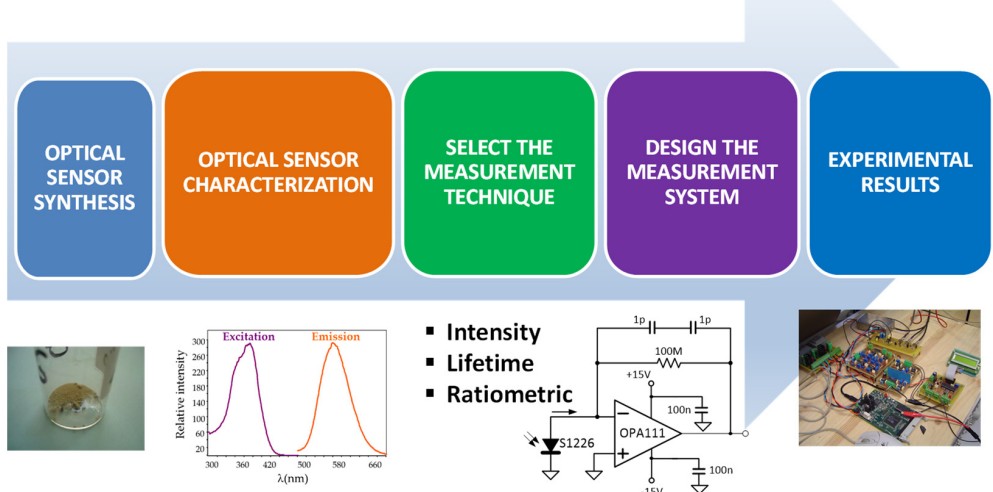

**Figure 13.** Design flow for an optoelectronic instrumentation system involving optical sensors.

### 4.1. Fiber Optic pH Sensor Based on Fluorescent Ratiometric Intensity Measurement

As is well known, pH is an important parameter in various fields of science and technology, such as environmental control, as well as in many industrial processes and in medicine, where physiological, rapid, and in situ pH measurements are sometimes needed. Historically, electrochemical sensors have been the most-used devices for pH measurements; however, these devices suffer from certain limitations, such as a lack of accuracy in measuring extreme pH values and difficulties in measuring aqueous suspensions of organic matter or low ionic resistance solutions. These problems limit the use of glass electrodes for certain applications. In these cases, optical sensors can be a good alternative, as they are less likely to suffer from the described problems.

#### 4.1.1. Optical Sensor Characterization

We prepared the active pH phase by entrapping the pH indicator in a sol–gel silica matrix during the polymerization process. Once the optical sensor was prepared, we obtained its excitation, emission, and reflection spectra by means of a fluorescent spectrometer for different pH values, as shown in Figure 14. From this figure, three important consequences can be observed that will affect the design of the system:

(1)　The pH value increases the emission of the fluorescent light in a quasilinear manner;

(2) The excitation and emission peaks of the sensor are close to each other ($\lambda_{\text{exc}}$ = 528 nm, $\lambda_{\text{emi}}$ = 549 nm). To avoid overlapping between both spectra, a light source with an emission peak below the sensing phase excitation maximum may be adequate;

(3) The excitation spectra depend on the pH and their intensity change in the same way as the emission spectra.

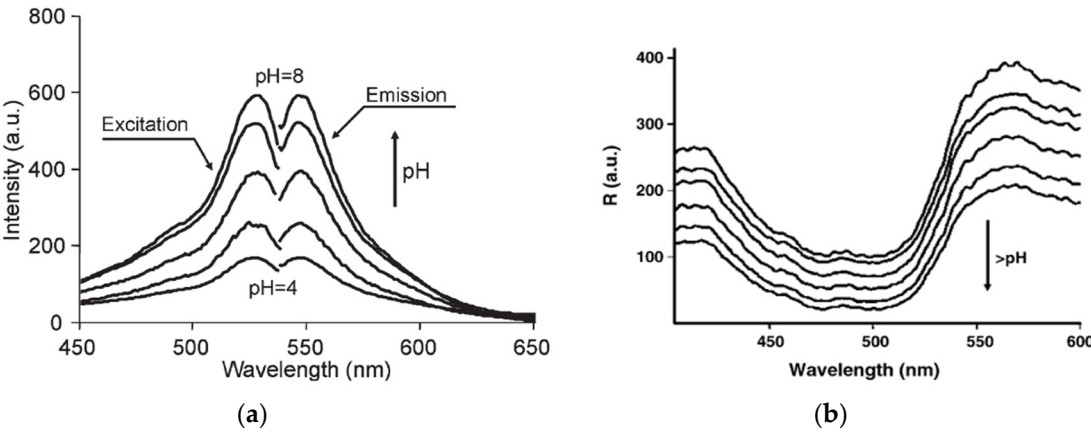

**Figure 14.** (**a**) Excitation and emission spectra of the sensing phase as a function of pH. (**b**) Reflectance spectra of the sensing phase as a function of pH.

### 4.1.2. Measurement Technique

As was explained in Section 3.2, the measurement of luminescence intensity is a simple technique, although its accuracy is compromised by variations in the excitation light, optical path, and concentration of the luminophore; therefore, ratiometric measurements are preferable in comparison to absolute-intensity-based measurements. In this way, many of the possible sources of error can be eliminated. During the optical sensor characterization, two types of characteristics were obtained: the emission spectra of the sensing phase as a function of pH and the reflectance spectra of the sensing phase as a function of pH. From these characteristics, a novel ratiometric intensity measurement technique was proposed based on the ratio between the fluorescent emission intensity and the reflected light intensity reaching the two photodetectors [48], with both signals being proportional to the perturbation that was to be eliminated.

The light that reaches the optoelectronic sensor is that reflected by the chemical sensor ($I_R$) and the one that corresponds to its fluorescent emission ($I_F$). The first of these has two components, specular reflection ($I_{SR}$) and diffused reflection ($I_{DR}$), both with the same wavelength as the incident signal but with a substantial difference: the specular reflectance does not depend on the pH because it does not interact with the sensor and only depends on the angle of incidence. The diffuse reflectance, on the other hand, does interact with the sensor and depends on the dispersion and absorption coefficients of the analyte, after which a pH function will result. Diffuse reflectance does not involve a process with loss of energy, so its wavelength remains the same as that of incident light but is reissued isotropically. The relationship between the intensity of the fluorescent emission ($I_F$) and the intensity of reflected light ($I_R$) that reaches the two photodetectors is:

$$R_{\text{t1}} = \frac{I_F}{I_R} \qquad (12)$$

The fluorescent emission of the sensor phase reached by the photodetector can be expressed as:

$$I_F = k_1 I_{\text{ex}}[\text{Ind}] \qquad (13)$$

where $k_1$ is the proportionality constant, $I_{\text{ex}}$ is the intensity of the excitation light reaching the detection phase, and [Ind] is the concentration of the fluorescent indicator. On the other

hand, the total light reflected by the sensor phase is the sum of the specular reflected light and the diffused reflected light:

$$I_R = I_{SR} + I_{DR} \tag{14}$$

The specular reflectance is independent of pH, due to the absence of any interaction with the sensor phase. This is a fraction of the reflected excitation light following Snell's laws, so it only depends on the incident angle:

$$I_{SR} = k_2 I_{ex} \tag{15}$$

where $k_2$ is the proportionality constant. The diffuse reflectance depends on pH through the concentration of the fluorescent form of the indicator:

$$I_{RD} = -k_3 I_{ex}[Ind] \tag{16}$$

where $k_3$ is the negative proportionality constant, since the diffuse reflectance is inversely proportional to the pH (Figure 2); therefore, Expression (14) is:

$$I_R = -k_2 I_{ex} - k_3 I_{ex}[Ind] \tag{17}$$

and Expression (12) is:

$$R_{t1} = \frac{k_1 I_{ex}[Ind]}{k_2 I_{ex} - k_3 I_{ex}[Ind]} = \frac{k_1[Ind]}{k_2 - k_3[Ind]} \tag{18}$$

As such, this ratiometric method is at least theoretically independent of drifts in excitation light. The main drawback of this method is the need to separate the two emissions, so it is a selective optical filtering method.

### 4.1.3. Measurement System

Figure 15 shows a block diagram of the proposed measurement system. The excitation source must meet the requirements established for the excitation spectrum of the chemical sensor and produce a signal that is unchanged over the life of the system. To reduce the effect of external interference (light from other sources), the light from the excitation source is modulated to a certain frequency. This frequency must not be so high as to compromise the dynamic limitations of the other components of the system or so low as to be confused with signals from artificial lighting or other sources. In this example, 1 kHz may be an adequate frequency. Interference can simply be eliminated by filtering the output signal to leave only the frequency corresponding to the modulation frequency of the excitation source.

The excitation source must be able to modulate light under these conditions. In our case, we chose a high-intensity blue LED (HLMP-CB30), the peak wavelength of which was 473 nm. This peak wavelength was slightly shifted from the sensing phase excitation maximum ($\lambda_{exc}$ = 528 nm) to avoid overlapping with its emission peak, which was 549 nm, as can be seen in Figure 16a. An alternative could be the use of a laser diode, although the particularities of their v-i characteristic curve means they are not the most suitable for light modulation. To generate the excitation wave, we used a scheme composed of a 1 kHz sine wave oscillator followed by a circuit to shift the continuous level and a current amplifier to provide the adequate excitation current level, as shown in Figure 16b. The light emitted by the sample passes through a narrow bandpass interference filter, which transmits specific wavelengths and blocks the rest. We selected the Edmund Optics NT43-074 interference filter, the central wavelength for which is 550 nm, the same as the peak of the emission spectrum. Because the level of fluorescence emitted is high, a photodiode with a spectral characteristic adapted to the emission spectrum of the chemical sensor is suitable. We chose a silicon photodiode (Hamamatsu S1226 silicon PD) with a peak wavelength at 550 nm. The conditioning circuit was based on a high-gain current-to-voltage converter. The converter

output signal is filtered using a fourth-order bandpass Butterworth filter, with the center frequency at 1 kHz, in order to eliminate interfering signals that may reach the detector. This signal is rectified by a precision rectifier. Low-frequency noise is eliminated by means of a 1 Hz low-pass filter. Finally, a gain and offset circuit is added to amplify the output signal and adjust the offset, which are not shown in Figure 16d.

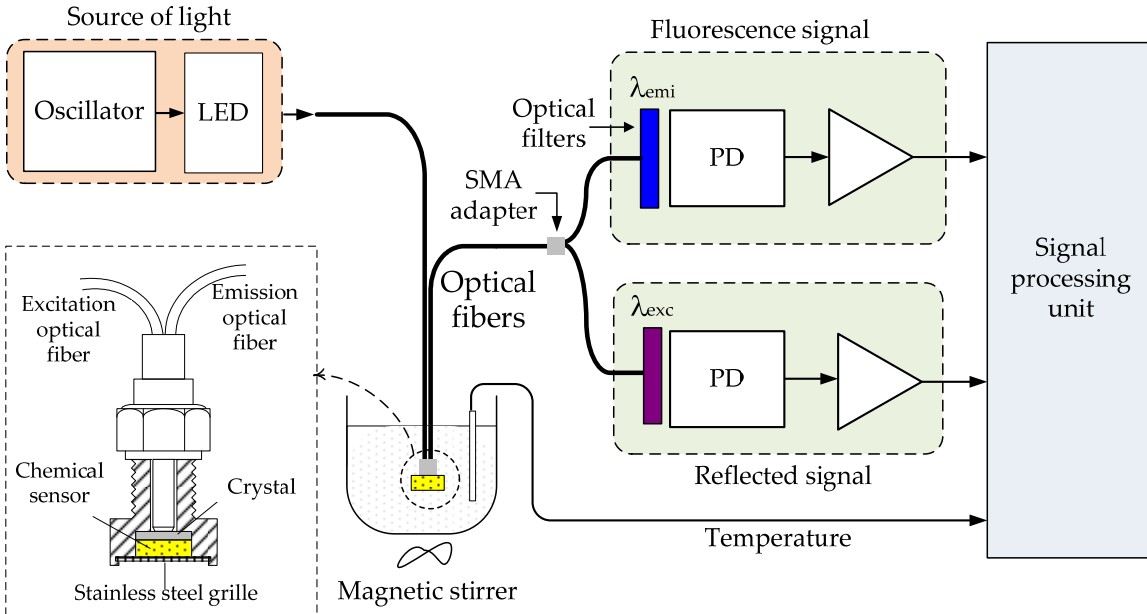

**Figure 15.** Simplified block diagram of a ratiometric fiber optic pH sensor used for intensity measurements.

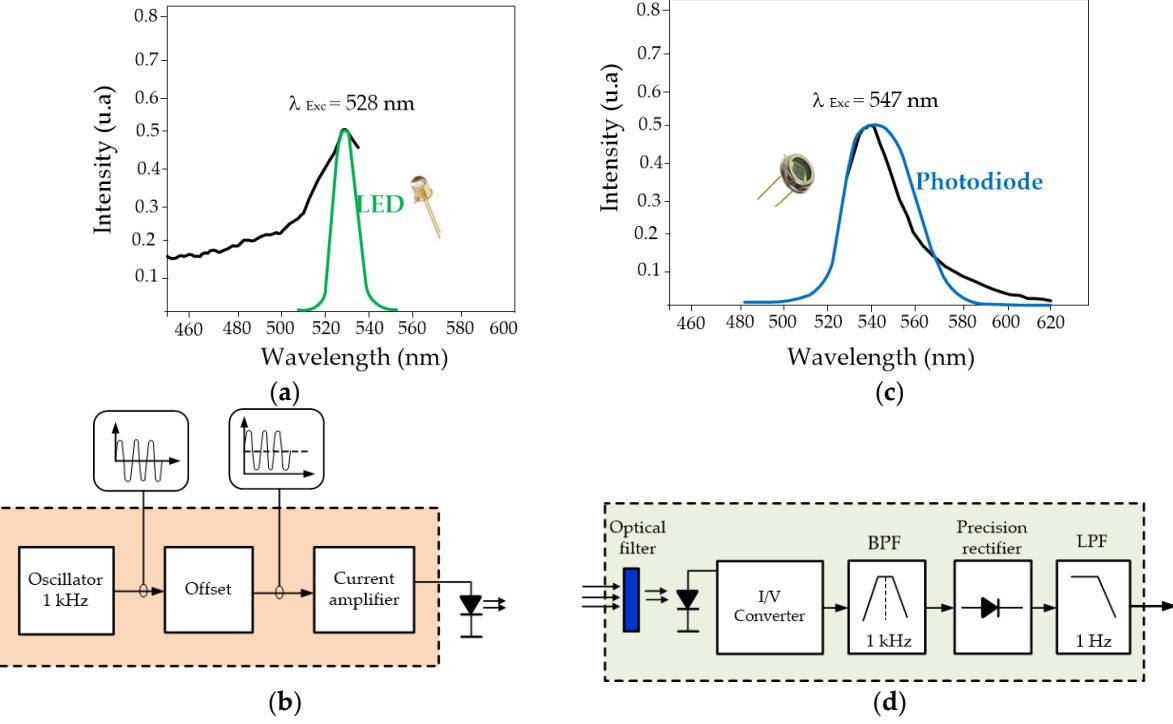

**Figure 16.** (**a**) Excitation spectrum of the chemical sensor and LED. (**b**) Optical sensor and photodiode emission spectra. (**c**) Block diagram of the light source. (**d**) Block diagram of the light detector.

The channel used for fluorescence detection uses an optical interference filter, with a through band centered at 550 nm, coinciding with the peak emission of the fluorescence. The channel for the detection of reflected light uses an optical interference filter with a passband centered at 470 nm, quite close to the wavelength of the emission peak of the excitation LED used. The filter setup is relatively complex because the normalized sizes are quite large (minimum diameter of one inch), meaning they have to be trimmed and inserted into the SMA connector.

The design of the measurement probe, although it does not require electronic circuits, is a fundamental element of the measurement system. The chemical sensor must be in contact with the medium. One way to avoid the dispersion of the chemical sensor is by using a grid or mesh that is thin enough, according to the granulometry of the sensing substance, to prevent it from dispersing and to allow the passage of the medium by contacting the sensor. It has been observed that the linearity and resolution are affected by the distance between the optical fiber and the sensor phase. In general, the greater the distance, the lower the upper limit of the linear margin of measurement and the lower the resolution. The light produced by the LED is brought to the detector by a plastic optical fiber measuring 1 mm in diameter, while the light emitted by the sensor is collected by an optical fiber measuring 1.5 mm in diameter, which is then bifurcated into two branches measuring 1 mm in diameter each, as shown in Figure 15. The reason for using a fiber with a greater diameter for the reception than for the rest of the fibers is to facilitate the collection of light. Doubling the fiber into two branches improves the results compared to using one excitation fiber and two receiving fibers—one to collect reflected light and the other to collect fluorescence light. A more detailed discussion of this approach can be found in [49].

### 4.2. Fiber Optic Oxygen Sensor Based on Phosphorescence Lifetime Measurement

Oxygen is undoubtedly one of the most important analytes on earth. Unfortunately, few sensors are able to measure concentrations of 100 nM or less [50,51]. In these cases, fiber optic sensors have proven to be indispensable tools for oxygen quantification, replacing the conventional Clark electrodes. Optical oxygen sensors are based on the attenuation of the luminosity of a phosphorescent indicator by the analyte. The latest generation of indicators is dominated by phosphorescent complexes with platinum group metals that have lifetimes of between a few microseconds and a few milliseconds.

### 4.2.1. Optical Sensor Characterization

The presence of dissolved oxygen frequently attenuates the light emission of a luminescent indicator because of the paramagnetic properties of molecular oxygen, which cause cross-system crossings and conversions of excited molecules to the triplet state, resulting in phosphorescent emission. For this chemical sensor, namely an Al-Ferron sensor with sol–gel support [52], the peak excitation is at 390 nm and the emission spectrum has a peak at 590 nm, as can be seen in Figure 17. The large separation between the two wavelengths simplifies the selection of the excitation source. On the other hand, a low level of phosphorescent emission is detected, which will also be attenuated using optical fibers, in turn informing the choice of photodetector.

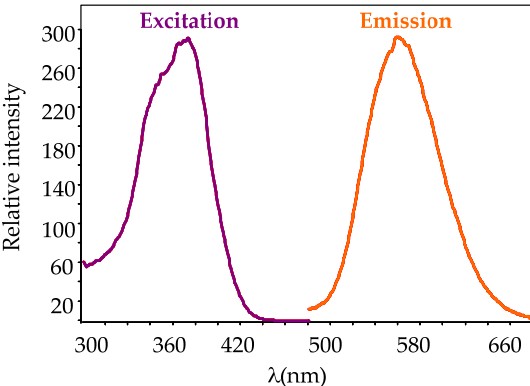

**Figure 17.** Spectrum of the Al-Ferron sensor in sol–gel support.

### 4.2.2. Measurement Method

The proposed method involves measuring the lifetimes of the phosphorescent emission, in which oxygen acts as an attenuator of the emission. The lifetimes are not affected by variations in the intensity of excitation light, in the amplitude of light emitted by the decomposition of the luminescent molecule, drifts in the gain of the detection system, or by optical interference in the optical path; therefore, it is not necessary to correct for the emitted light as a function of the excitation light. In addition, the system is insensitive to dirt in the optical elements. For the measurement of phosphorescence lifetimes, the sensor is excited with a pulse of high-intensity light and at a certain wavelength. The sensor emits light at another wavelength, the intensity and lifetime of which depends on the concentration of the substance being measured (in this case the concentration of oxygen is being measured). The sensor acts by attenuating the emission of the luminophore, as we noted in Section 2.

### 4.2.3. Measurement System

The proposed measurement scheme is shown in Figure 18 [53]. To excite the chemical sensor requires a pulse of high-intensity light. We chose a UV LED with an emission peak close to that of the excitation of the chemical sensor.

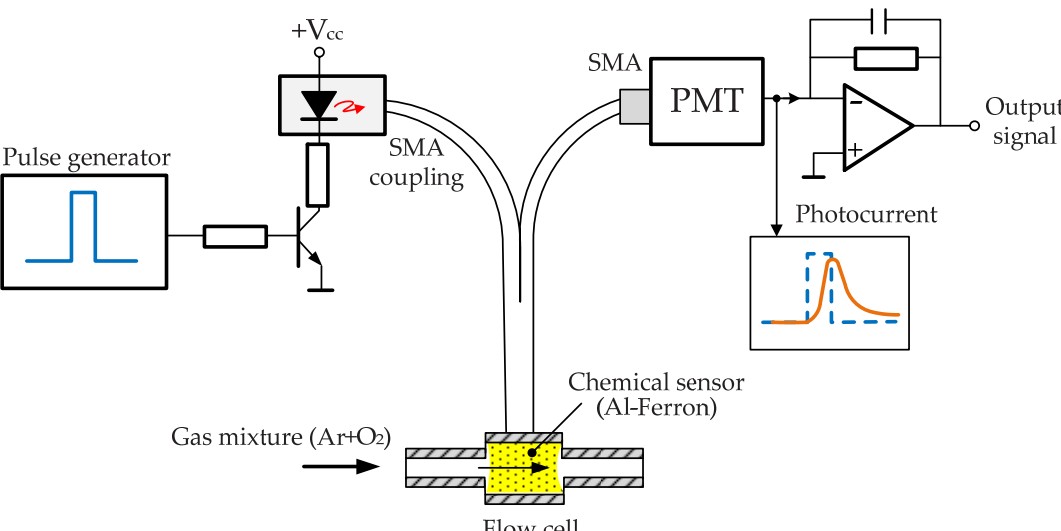

**Figure 18.** Design of an optical oxygen sensor based on phosphorescence lifetime measurements.

The phosphorescence emitted by the sensor is of low intensity. Specifically, the power of the light collected at the end of the fiber is less than 10 pW. Under these conditions, the best option is a photomultiplier tube (PMT), given its high sensitivity and good S/N ratio

for low-light applications. It is possible to increase the sensitivity of the PMT by increasing the supply voltage, although other negative effects can occur, such as an increase in the dark current, which results in an offset voltage at the output. The output current of the PMT is converted into voltage in the first analog stage. Before making the measurement, it is necessary to introduce a certain delay to avoid the presence of residual luminescence. Once this delay is introduced, the curves in Figure 19a are obtained, which represent the attenuation of phosphorescence due to different concentrations of oxygen.

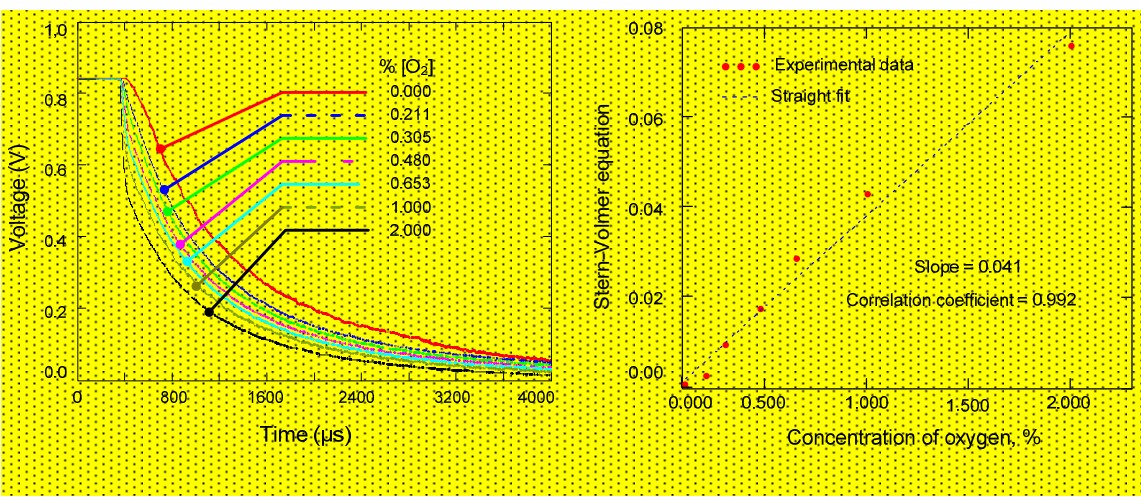

**Figure 19.** (**a**) Phosphorescent responses for different oxygen concentrations. (**b**) Calibration curves according to lifetime lengths.

To extract the lifetimes from these curves, we adjusted the curves first using simple exponentials and second using a double exponential. The latter setting is better but requires more processing. Once the lifetimes are known, the concentration of the analyte can be calculated using the well-known Stern–Volmer equation, which linearly relates lifetimes to oxygen concentrations:

$$\frac{\tau_0}{\tau} = 1 + K_{SV}\,[O_2] \tag{19}$$

where $\tau_0$ and $\tau$ are the lifetimes in the absence and presence of oxygen, respectively. $K_{SV}$ is the Stern–Volmer constant, which is proportional to $k_d$, the biomolecular quenching constant, and $\tau_0$. Finally, Figure 19b shows the calibration curve according to the lifetime of the phosphorimeter using the Al-Ferron chemical sensor for low oxygen concentrations.

## 5. Conclusions

In the field of analytical chemistry, the positive development of optical sensors has been unquestionable. Such sensors are successfully applied for the measurement of oxygen, pH, hydrocarbons, carbon dioxide, acetone, and organic vapors, among others. The success of this type of sensor has been accompanied by the parallel development of optoelectronic instrumentation. One of the elements that has contributed to the surge in applications is the emergence of low-cost, high-performance optical devices. For most optical sensors luminescence intensity is the simplest technique to measure the analyte concentration; however, this technique introduces errors from the excitation light, optical path, and concentration characteristics, among others. An alternative to measuring the absolute intensity is to use a ratiometric intensity measurement. This technique is relatively simple and can mitigate variations in the indicator concentration, geometry, and source intensity. Another alternative is to measure the luminescence lifetimes, either in the frequency domain or in the time domain. The first approach is mostly applied to indicators with lifetimes in the µs or ms range. The second approach requires more complex and expensive instrumentation, allowing the elimination of background fluorescence, which usually

decays within 100 ms. In this article, we have presented the measurement principles for this kind of instrumentation by means of two representative examples.

**Author Contributions:** Conceptualization, F.F.M. and J.M.C.-F.; methodology, J.M.C.-F. and A.S.C.; software, A.L.M.; validation, M.V.L., A.S.C., and M.T.F.-A.; formal analysis, M.V.L.; investigation, M.T.F.-A. and M.V.L.; resources, J.C.C.R.; data curation, J.C.C.R.; writing—original draft preparation, F.F.M.; writing—review and editing, F.F.M.; visualization, F.F.M.; supervision, J.M.C.-F.; project administration, A.L.M.; funding acquisition, J.M.C.-F. All authors have read and agreed to the published version of the manuscript.

**Funding:** This research received no external funding.

**Conflicts of Interest:** The authors declare no conflict of interest.

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
