# Peer review of "Optoelectronic Instrumentation and Measurement Strategies for Optical Chemical (Bio)Sensing"

_applsci, doi:10.3390/app11177849_

Round 1

Reviewer 1 Report

In general, the paper is well written and provides comprehensive details about the optical chemical instrumentation. However, there are some points I would like to raise to improve the quality of the manuscript. Please see the attachment for the comments.

Reviewer 2 Report

In this paper, the authors focus on the development of low-cost optoelectronic instrumentation and measurement strategies for optical chemical (bio)sensing. This article is clear, concise, and suitable for the scope of the journal. Several small suggestions are supplied:

  1. Suggest the authors improve the introduction part with more related references. 
  2. Suggest the author improve the graph of Monochromators in table 3.
  3. Suggest the author improve the format of Figure 3 and Figure 17.
  4. Suggest the author check the references carefully, some formats should be modified.
